# Results of a Clinical Trial Showing Changes to the Faecal Microbiome in Racing Thoroughbreds after Feeding a Nutritional Supplement

**DOI:** 10.3390/vetsci10010027

**Published:** 2022-12-30

**Authors:** Vicki J. Adams, Neil LeBlanc, Johanna Penell

**Affiliations:** 1Vet Epi, Abbey Farm Cottage, Ixworth, Suffolk IP31 2JP, UK; 2Consultant Norrlandsgatan 36D, 75229 Uppsala, Sweden; 3Department of Clinical Sciences, Faculty of Veterinary Medicine and Animal Science, Swedish University of Agricultural Sciences, 75007 Uppsala, Sweden

**Keywords:** faecal microbiota, thoroughbreds, training, nutritional supplement, clinical trial

## Abstract

**Simple Summary:**

Bacteria within the gastrointestinal tract communicate with the immune system and the brain, have a role in energy metabolism and are responsible for gastrointestinal health and gut wall integrity. Veterinary research has reported links between gut bacteria and health. The so-called good bacteria can maintain health while proliferation of pathogenic bacteria can occur in certain conditions, causing colitis, inflammation, colic and dysbiosis. Genetic sequencing of gut bacteria can be used to provide a picture of the composition of the gut microbiome. This information can be used to manage the diet to provide an environment where the good gut bacteria can thrive. This study showed that feeding a prebiotic nutritional supplement resulted in mean and median percent decreases in *Bacteroidetes*, increases in *Firmicutes* and the *Firmicutes:Bacteroidetes* ratio that were significantly greater than zero for the treated horses only. Supplemented horses (8/10) were more likely than control horses (2/10) to show an increase in *Firmicutes* of a ≥9% with ≥24% increase in *Clostridia*, ≥5% decrease in *Bacteroidetes*, ≥16% increase in the F:B ratio and ≥2% increase in *Actinobacteria* (*p* = 0.01). This provides useful information for further investigations on long-term effects on the microbiome and on health and racing-related outcomes.

**Abstract:**

Next-generation sequencing (NGS) has been used to evaluate the effect of various interventions on the equine microbiome. The aim of this randomised blinded clinical trial was to determine if a prebiotic nutritional supplement would result in a change from baseline in the faecal microbiome composition of racing Thoroughbred horses in training being fed a high concentrate/grain-based diet to be more similar to that found in forage fed/pasture grazed horses. Thirty-two horses on one training yard were randomised to either receive the supplement or not. Faecal samples were collected at baseline, 6 and 12 weeks for NGS of the 16S ribosomal subunit gene. Twenty-two horses completed the trial, met the inclusion criteria and were included in the intention to treat analysis; 20 horses were included in the per protocol analysis. The mean and median percent decreases in *Bacteroidetes*, increases in *Firmicutes* and the *Firmicutes:Bacteroidetes* ratio were significantly greater than zero for the treated horses only. Supplemented horses (8/10) were more likely than control horses (2/10) to show an increase in *Firmicutes* of a ≥9% with ≥24% increase in *Clostridia*, ≥5% decrease in *Bacteroidetes*, ≥16% increase in the F:B ratio and ≥2% increase in *Actinobacteria* (RR = 4, 95% CI: 1.1–14.4, *p* = 0.01). This provides useful information for further investigations on long-term effects on the microbiome and on health and racing-related outcomes.

## 1. Introduction

The equine gastrointestinal tract (GIT) houses a complex and diverse microbial population that has been described in several studies using next generation sequencing [1,2,3]. Changes in the microbiome are usually the result of changes in the diet, especially abrupt changes, refs. [2,3,4,5] or environment, including factors such as stress, injury, disease and antibiotic or other drug use [6,7]. A study presented differences in the mean relative abundances of specific bacterial phyla in forage fed/pasture grazed horses compared to those with an intestinal disease such as colitis [1]. Studies have also shown that the microbiome is different among healthy horses, suggesting that each horse’s microbiome may be unique and there is considerable variation between horses (Appendix A) [1,3,8]. Whilst both probiotic and prebiotic supplements have been commercially available for horses for a considerable time, there is a lack of evidence on the effect of prebiotics on the microbiome. Prebiotics are soluble plant fibres that cannot be digested by enzymes in the stomach or small intestine. This means they reach the hindgut where they are broken down by specific microbes and function as substrates to stimulate the growth of beneficial microbes. Prebiotics are used to provide nutrition for the equine microflora with the intention of improving the balance of these microorganisms. Studies have also shown that while the microbiome varies along the intestinal tract, faecal samples can be used as an indication of the microbiome of the right dorsal colon [4,5]. 

The purpose of the current study is to report the results of a pilot study and a randomised blinded clinical trial to investigate the effect of daily feeding of a prebiotic nutritional supplement on the microbial composition of faecal samples as a proof-of-concept study to aid in the design of a future longitudinal study. The aim was to determine if the supplement would result in a change from baseline in the faecal microbiome composition of racing TBs in training being fed a high concentrate/grain-based diet to be more similar to that found in hforage fed/pasture grazed horses (Appendix A) [1,3,8]. We hypothesised that a greater proportion of treated horses would show changes in the faecal microbiome compared to control horses. A secondary aim was to document the baseline faecal microbiome of racing TBs in training. 

## 2. Materials and Methods

### 2.1. Pilot Study

Four TB racehorses in training were sampled at baseline and after 1 and 2 months of supplementation (see Section 2.4) and the samples sent to the testing laboratory (see Section 2.6 and Section 2.7). The observed changes in the Phylum Bacteroidetes, Firmicutes and the ratio of Firmicutes to Bacteroidetes (F:B) are reported (Table 1) and were used to estimate required sample size for the clinical trial.

### 2.2. Sample Size for Clinical Trial

A sample size estimate was derived from the results of the pilot study with the expectation that a similar effect of treatment would be apparent in the treatment group as in the pilot study. It was anticipated that some control horses would show changes and some treated horses would not show changes to the microbiome. Based on this, we anticipated that up to 30% of control horses and at least 70% of treated horses would show similar changes to those seen in the pilot study; this resulted in an estimated 24 horses that would be needed using a 1:1 ratio of treated to control horses with at least 80% power (beta = 0.20) to be able to detect a difference with at least 95% confidence (alpha = 0.05) in the proportion of treated horses that would experience similar changes to the faecal microbiome specified as detecting a relative risk of >1.5. We planned to recruit and enroll a total of 32 horses to the study to allow for withdrawals or loss to follow-up or if the efficacy of the supplement was not as high as anticipated. Ethical approval for this trial was obtained from the RCVS Ethics Review Panel (ERP reference 2019/48/Adams).

### 2.3. Subject Recruitment and Study Protocol

Horses were eligible for inclusion if they were TB horses in full training throughout the study period on the participating yards in Newmarket, UK. Horses would be excluded from the study if antibiotics were required during the study, if a horse experienced any physical injury that resulted in a disruption of training, if a horse required other treatment for an illness or injury that required training to be stopped or if any horse refused to eat the supplement. 

The original study protocol included a plan to recruit 16 horses each from two training yards registered with the same veterinary practice for routine care. However, for the practical reasons of efficiently collecting samples, a decision was made to only use one yard to enroll 32 horses; a side benefit of this approach was that it would help to minimise the variability associated with using different training yards. Horses were randomised to either the treatment (prebiotic nutritional supplement) group or the control group (no treatment) after a baseline faecal sample was obtained. The horses were randomised in two blocks of 16 horses (Figure 1). Each block of 16 horses were housed in sequentially numbered box stalls and were randomised by splitting each set of 16 box stalls into two groups of eight sequentially numbered box stalls, one of which became the treatment group and the other the control group. Each of the two blocks of 8 horses in the treatment group were fed separately to facilitate feeding of the supplement, to minimise cross contamination of the control groups and to allow blinded faecal collection. The intended treatment had a very distinctive odour due to the ingredients, making a placebo control supplement unfeasible. There were measures in place to ensure that those feeding the horses did not reveal which horses were in which groups to anyone else involved in the study. The horses in the treatment group received one heaped scoop of the pelleted nutritional supplement (~10 g) once daily for 12 weeks. 

Most of the trial horses were housed in the initial two groups of 16 box stalls throughout the study although there was some movement of horses within the yard. If a horse was moved from the original box stall to a different box stall during training, sampling was continued by the PI who was blinded to treatment group. Any new horse that was then placed in the vacated box stall was added to the study and was also sampled from that point on to the end of the study to maintain blinding of treatment group assignment. Whether any of these additional horses were added to the study was decided by the PI’s assistant (RA) when the two groups were partially unblinded for statistical analysis.

### 2.4. Nutritional Supplement

The prebiotic nutritional supplement was a pelleted proprietary formulation containing the following ingredients in descending order of amount included: Wheatfeed Meal, Micronised Beet Pulp, Oatmeal, Hempseed Meal, Garlic, Cinnamon, Common Thyme, Peppermint, Fennel, Cleavers, Nettle, Dicalcium Phosphate, Molasses, Slippery Elm, Quassia, Elecampane, Sunflower Oil, Seaweed Meal, Vitamin C (Verm-X, Paddocks Farm Partnership Ltd., Taunton, UK).

### 2.5. Blinding

Blinding was used to withhold information about the assigned interventions from those involved in the trial who may potentially be influenced by this knowledge, including the laboratory personnel doing the faecal microbiome analysis and the principal investigator (PI) carrying out the faecal sampling and data analysis (lead author VJA). Each faecal sample was identified by a unique code. The horse and sample numbers were known only to the PI’s assistant (RA) who oversaw the random assignment of treatment group by block once baseline sampling was completed. The PI remained blinded until the data collection and sequencing of samples were completed; the two groups of horses were revealed to the PI by the assistant using coded initials (simply W or X) to allow for statistical analysis. Identification of the treatment and control groups only occurred once the statistical tests were completed to allow the results to be written up. The participating training yard had its own documented training and feeding regimes that were followed throughout the study and any variations in diet would be reported to the PI. This included a record of all medications given. Whether the horses raced or not during the study period was recorded from information on the Racing Post website (www.racingpost.com [accessed 28 September 2022]). 

### 2.6. Faecal Sampling

A total of 3 faecal samples were collected from the trial horses between January and May 2020: at baseline, after 6 weeks and at 12 weeks when the trial ended. Each sampling session took place in the early afternoon from approximately 2 p.m. to 3:30 p.m. after the horses had finished their daily exercise and were back in their clean box stalls. A fresh warm spontaneously produced faecal sample was collected off the ground by the PI who wore a new pair of nitrile blue accelerator free sterile medical gloves for each sample. Approximately 100 mg of faeces were collected from the middle of a faecal ball using sterile plastic tweezers and avoiding collection of faecal material that had contact with the ground. Samples were placed in a sterile Cryovial 2.0 mL tube (certified DNase-free, RNase-free, DNA-free and Pyrogen-free) to which ~1.5 mL 96% ethanol was added to completely fill the tube. Each batch of samples were refrigerated before posting to the laboratory for DNA extraction and sequencing. Each faecal sample was identified using a unique code to maintain blinding.

### 2.7. Laboratory Analysis

Upon arrival at the laboratory, each batch of samples were frozen at −80 °C until DNA extraction and sequencing could be performed. Next generation sequencing was performed for 16S ribosomal subunit gene (16S rDNA) sequencing at a commercial laboratory in the UK. Bacterial DNA was extracted from the faecal samples using the Qiagen PowerFecal 96 kit as per the manufacturer’s instructions. Libraries were prepared using the standard Illumina 16S metagenomics protocol and Nextera v2 indices. Samples were quantified via Qubit fluorescence spectrophotometry and pooled at equimolar ratios. As per the Illumina MiSeq protocol, the pooled samples were diluted to 4pM for loading onto a MiSeq v2 platform and sequenced in 2 × 300 bp format together with 10% PhiX control spike. Resulting FASTQ data were uploaded to Illumina BaseSpace cloud computing environment and analysed using the 16S Metagenomics app to assign reads to bacterial operational taxonomic units (OTUs). Any samples with less than 30,000 reads or that were considered failures were excluded from the study. After all samples were processed and analysed at the laboratory, the uploaded data from sequencing were downloaded and entered onto a spreadsheet using the coded sample identification numbers which were associated with each randomly assigned horse number. The PI was blinded to treatment group throughout data entry and outcome coding. Once the sequencing results were added, the data were sorted by horse number and then by sample number (1, 2 and 3) to allow blinded assignment of changes in bacterial abundance and outcome for each horse. The relative abundances of individual bacteria are reported as a percent of the total abundance and the Shannon species diversity (SSD) index is reported as a measure of the entropy of Species-level classifications in the sample (alpha diversity [9,10]). The PI’s assistant had access to the horse numbers for the horses that were enrolled into the clinical trial and randomised to treatment group and used this to create a list of horse numbers for analysis and outcome evaluation. 

### 2.8. Statistical Analysis

Partial unblinding of the groups was done when the PI’s assistant assigned either the letter W or X to signify which group each horse belonged to, without revealing which of these was the control and treatment groups, for statistical hypothesis testing. Since some of the data followed an approximately normal distribution with equal variances and some did not, both parametric and non-parametric tests were used. Two-sample T tests were used to compare the mean values of the two groups at baseline and at 12 weeks for the following variables: SSD index, number of species, F:B and relative abundances of 12 phyla (*Firmicutes, Bacteroidetes, Spirochaetes, Fibrobacteria, Proteobacteria, Tenericutes, Verrucomicrobia, Acidobacteria, Candidatus, Actinobacteria, Synergistes and Fusobacteria*) as well as 2 classes (*Clostridia and Erysiplotrichia*). Two-sample T tests were also used to compare the mean percent changes from baseline to the 12-week sample for the same variables and these tests were repeated using non-parametric median tests. One-sample T tests and Wilcoxon signed rank tests were used to evaluate whether the mean percent changes from baseline were significantly different from zero for each treatment group. 

This was followed by outcome assessment where outcome was defined as a change in the microbiome and expressed as a dichotomous variable (yes/no). Two definitions were used for overall change based on the magnitude of change in relative abundances of specific groups of bacteria: one that included five criteria and one that had a sixth criterion. The first definition of change was based on a previous study reporting proportions of these individual variables for forage fed versus colitis horses, with the direction of change from colitis to forage fed used as the desirable direction, along with a magnitude of change for each variable, and defined as meeting the following 5 criteria:(1)Increase in the phylum *Firmicutes* (F) of ≥9%;(2)Increase in the class *Clostridia* of ≥24%;(3)Decrease in the phylum *Bacteroidetes* (B) of ≥5%;(4)Increase in the F:B of ≥16%;(5)Increase in the phylum *Actinobacteria* of ≥2%.

The second definition of change was defined as meeting the above 5 criteria as well as a decrease in the class *Erysipelotrichia* of ≥8%. Final unblinding of the control and treatment groups only took place once the statistical tests were completed. Using these criteria, crosstabulations and Fisher’s exact tests were used to compare the proportions of the two groups (W and X) showing the defined changes in relative bacterial abundances from baseline to the 12-week sample. The relative risk (RR), its standard error and 95% confidence interval (CI) were calculated according to Altman (1991) [11]. Logistic regression analysis was also used to evaluate the effect of potential confounding variables including sex, whether they were gelded during the trial, treatment with a medication (e.g., phenlybutazone or omeprazole) and whether horses had raced before the trial. Results are reported as relative risk (RR) with 95% confidence interval (CI). The level of significance was set at *p* < 0.05. In order to examine the optimal potential benefits and also ensure bias was minimalised both per protocol and intention to treat analyses were performed. A per protocol analysis was performed on the horses that met the inclusion criteria and completed the trial with adherence to the study protocol and an intention to treat analysis was also performed based on randomisation to treatment or control group.

## 3. Results

### 3.1. Pilot Study

In the pilot study of 4 Thoroughbred (TB) racehorses, the faecal microbiome composition at baseline was dominated by the Phylum Bacteroidetes followed by *Firmicutes* with a mean and median F:B of 1.16 that ranged from 0.67 to 1.63 (Table 1). After two months of supplementation, the mean relative abundance of *Bacteroidetes* fell from 32 to 21% (a 34% decrease) and the *Firmicutes* increased from 33 to 47% (a 42% increase), resulting in an increased F:B ratio that ranged from 1.3 to 7.3 (>400% increase) after supplementation for 2 months (Table 1, Appendix A).

### 3.2. Clinical Trial Subjects

After excluding seven horses that were either sold, bred, underwent colic surgery or were not on the yard at the time of sampling and including two additional horses that replaced two of the excluded control horses, a total of 27 horses had three samples collected (Figure 1). Three horses received antibiotics during the trial and were excluded from analysis, leaving 24 horses with three samples collected during the 12-week sampling period. Sequencing was successful for all but three samples: if samples were available from first and third collected the horse was included (*n* = 1) whereas if a horse had only first and second (*n* = 1) or second and third (*n* = 1) samples they were excluded. Outcome was classified for 22 horses that had first and last samples successfully sequenced. 

The 22 horses were of similar age and all were classified as 3 year-olds in training with a date of birth at the start of the study that ranged from 21 January 2017 to 6 May 2017 (Appendix A). There were three fillies in the control group and five in the treated group; there was one colt in each group that was gelded during the trial. All horses were fed one of two types of concentrate feed (Connolly’s RED MILLS Horse Care 10 Mix or 14 Mix, Goresbridge, Ireland), a low starch, high fibre, oat-free mix with added RED MILLS Nutrition Care package. Horses were also offered hay and haylage and there were no major changes in feeding during the trial. A total of eight horses received medication during the trial: two control and four treated horses were given phenylbutazone, two control horses were given omeprazole (Gastrogard, Boehringer Ingelheim, Germany, Appendix A). Seven control horses raced at least once prior to the trial in 2019 (one also raced in 2018) but did not resume racing until the trial was completed and three control horses did not start racing until the trial was completed (Appendix A). Three treated horses had raced at least once prior to the baseline sampling date but did not resume racing until after the trial was completed and the other nine treated horses did not start racing until after the trial was completed. 

Four horses were moved to a different box stall at least once during the trial and these horses continued to be sampled. Two of the horses that moved box stalls after the second sample was taken were control horses and these remained in the trial. The other two horses had been randomised to the treatment group and it was not clear whether these two horses did continue to be fed the supplement after moving so these two horses were excluded from the per protocol analysis (*n* = 20) but were included in the intention to treat analysis (*n* = 22).

### 3.3. Baseline Microbiome for Treatment and Control Groups

At baseline, unclassified reads made up between 3.1% to 6.5% of the total phyla for 20 horses. There were 10 phyla with a relative abundance of >0.1% in all horses at baseline and these included: *Firmicutes > Bacteroidetes > Spirochaetes > Fibrobacteria > Proteobacteria > Tenericutes > Verrucomicrobia > Acidobacteria > Candidatus > Actinobacteria*. The *Firmicutes* and *Bacteroidetes* dominated (Table 2, Figure 2), accounting for > 50% of all phyla and ≥70% of the total microbiome in 9 horses. The horses in the clinical trial had higher levels of *Firmicutes* and lower levels of *Bacteroidetes* at baseline compared to the horses in the pilot study. Independent sample tests comparing groups showed that the treated group had a significantly greater mean number of species at baseline compared to the control group (*p* = 0.009) although the medians were not significantly different (*p* = 0.18, Table 2). The only significant difference between the treatment and control groups in relative abundance of bacteria was that the treatment group had significantly greater mean relative abundance of the Phylum *Actinobacteria* (*p* = 0.04) although the medians were not significantly different (*p* = 0.18, Table 2). 

### 3.4. Microbiome Composition at 12 Weeks and Percent Change in Composition from Baseline for Treatment and Control Groups

After 12 weeks of supplementation, the treated group had a significantly lower median relative abundance of *Bacteroidetes* compared to the control group (*p* = 0.02) although the means were not significantly different (*p* = 0.08, Table 2). One-sample tests showed that the mean and median percent decreases in *Bacteroidetes* of 25% in the treated group was significantly different from zero (*p* = 0.007 and 0.013, respectively); this was also true for the mean increase of 23% and median increase of 20% in *Firmicutes* (*p* = 0.006 and 0.007, respectively) and the mean increase of 85% and median increase of 71% in the F:B (*p* = 0.013 for both). The percent changes of these variables in the control group were not significantly different from zero.

Independent two-sample tests comparing groups showed that there were no significant differences between groups in the mean or median number of species at the 12-week sample (*p* = 0.4 and 1.0, respectively, Table 2) or mean or median relative abundance of *Actinobacteria* (*p* = 0.06 and 0.18, respectively, Table 2). However, every horse in the control group showed an increase in the number of species at the 12-week sample while only 5 of 10 horses in the treatment group showed an increase. The percent increase in the number of species was significantly higher for the control group compared to the treatment group (*p* = 0.03 for mean and 0.02 for median). The mean and median percent increases in number of species was significantly different from zero for the control horses (*p* < 0.001 and *p* = 0.005, respectively) while this was not true for the treated horses.

### 3.5. Outcome for 20 Horses on a Per Protocol Analysis

A total of 20 horses completed the clinical trial with adherence to the randomised treatment assignment. Using the first definition for change in 5 criteria, significantly more treated horses (n = 8) showed these changes compared to control horses (n = 2) with a RR of 4.0 (95% CI: 1.1–14.4, *p* = 0.012, Table 3). Using the second definition for change in 6 criteria, significantly more treated horses (n = 8) showed these changes compared to control horses (n = 0) with a RR of 17.0 (95% CI: 1.1–259.9, *p* = 0.0004, Table 3). Logistic regression did not show any effects of sex, whether the horse was gelded during the trial, the use of medications or whether they had raced prior to the trial on any of the outcome measures (data not presented). 

### 3.6. Intention to Treat Analysis for 22 Horses

Including the two treated horses that were moved from their original box stall after the second sample was taken in an intention to treat analysis did not alter the results as these two horses still showed an improvement in the microbiome. Using the first definition for change in 5 criteria, significantly more treated horses (n = 10) showed these changes compared to control horses (n = 2) with a RR of 4.2 (*p* = 0.008). Using the second definition for change in 6 criteria, significantly more treated horses (n = 8) showed these changes compared to control horses (n = 0) with a RR of 14.4 (*p* = 0.002).

Stacked bar charts for the mean relative abundances at each sampling time by group and for each horse within treatment group are included as supplemental information. These show both the within and between horse variability in relative abundances and illustrate why we used each horse as its own control to define changes from baseline (Appendix A).

## 4. Discussion

This randomised blinded clinical trial in racing TBs showed that a prebiotic nutritional supplement resulted in a change to the faecal microbiome in a greater proportion of treated horses compared to control horses. Specific criteria for assessing improvement were applied before knowing the actual treatment group status to facilitate as objective evaluation as possible of the supplement on the composition of the faecal microbiome. We found that some of the treated horses did not show improvement of their microbiome despite being treated and some non-treated horses did show improvement. This is expected in a clinical trial and was the reason for randomisation and blinding. By limiting our analysis to consideration of the dominant phyla and two classes of bacteria, we were able to show that the treated horses that received the prebiotic nutritional supplement group were significantly more likely to show changes in the faecal microbiome compared to the control horses that did not receive the supplement. This study was also intended as a proof-of-concept study and the results have highlighted factors that need to be addressed in the design of a future larger longitudinal study.

The importance of the intestinal microbiota has been recognised and numerous studies have reported on the composition and changes in microbial abundance in relation to various factors. A recent review outlined the many factors that influence the equine GI microbiome, including nutrition and management factors (diet, supplementation [pre and probiotics], exercise, seasonal, spatial and social interactions), medications, animal-related factors (age, disease [colitis, diarrhoea, colic, laminitis] and stress (transportation, racing) [7]. Even so, there continues to be a lack of information on what comprises a ‘healthy’ or ‘normal’ faecal microbiome in horses and which criteria should be used to evaluate improvement after an intervention. Variability in equine gut microbiota composition among various breeds and age groups, within different gut compartments, as well as individual variability makes it difficult to generalise findings from previous studies. One study of 6 healthy Irish TBs [8] showed great variability in the relative abundance of bacterial phyla as does another study that included 6 horses of different breeds and ages [1]. The current study design attempted to overcome some of the limitations of previously published studies that have included the use of horses of different breeds, ages and under different management conditions. The fact that the current study took place on one training yard with horses of one breed, similar age and fed and managed in similar circumstances should minimise the likelihood of bias due to these potential confounding factors. While a crossover study would have helped to address the issue of within and between horse variability, very little is known about the persistence of the effect of the supplement and this made it difficult to define an adequate wash out period for such a study.

While digestive supplements that include probiotics and/or prebiotics have been used extensively to moderate the microbiome, both in vivo and in vitro studies show conflicting results [7]. According to FAO/WHO, probiotics are defined as “live microorganisms which when administered in adequate amounts confer a health benefit on the host” [12]. The supplement used in the current trial can be classified as a type of prebiotic given that a prebiotic has been defined as “a substrate that is selectively used by host microorganisms conferring a health benefit” [13]. Prebiotics include oligosaccharides and dietary fibre as well as polyphenols and fatty acids. Short-chain fructooligosaccharides have been shown to improve digestibility in senior horses [14] and have reduced the microbial disruption in the hindgut associated with sudden ingestion of barley (starch overload) [15]. 

Limitations to the current study include moderate sample size, limited follow-up time, lack of metabolite profiling and the large between-horse variability in microbiome composition. Differences in methodology of DNA extraction, library preparation protocols and sequencing platforms also make comparisons between studies challenging. A strength of the current study, in addition to the use of randomisation and blinding, was the use of one laboratory protocol for all samples. The Illumina based targeted 16S rRNA gene amplicon sequencing used in the current study provides a community census where the data are compared to databases to report relative abundance of detected taxa. Quantification of microbial taxa requires different methods and because metagenomics alone cannot identify the functional pathways of microbial communities, future studies need to include non-genomic analyses such as proteomics and metabolomics to investigate associations between alternations in relative bacterial abundances, fermentation end products and physiological changes and health/performance outcomes. 

The pilot study baseline samples were collected in the summer (June 2018) and the microbiome showed greater similarity to that of the horses with colitis in the study by Costa and others than the trial horses did at baseline [1]. The pilot study samples after supplementation (collected in August 2018) also showed greater changes in the most common phyla (Firmicutes and Bacteroidetes) than the treated horses in the current study. Others have shown that there are seasonal variations in the faecal microbiome of horses maintained on pasture over a 12-month period with minimum changes to their management [16]. Additionally, a recent study has shown differences in relative abundances of bacteria between summer and winter seasons of sampling [17]. While seasonal differences may be related to pasture access, a different pattern of training and/or racing between summer and winter may have affected the results in our study. The pilot study horses also showed less variability in the relative abundances of bacteria compared to the clinical trial horses. 

The effect of storage in a refrigerator before transport by post may have affected the results. Given that cold-chain storage is not always feasible for samples collected in the field and that thawing of samples during transport may significantly reduce DNA integrity and influence the accuracy of downstream microbiome analysis [18,19,20,21], several studies have aimed to assess the effect of time and temperature on microbial communities in faecal samples [19,21,22,23,24,25,26,27]. The methods used were variable and the results are conflicting. The uniqueness of microbiota composition and influencing factors in different settings, along with lack of standardisation of study procedures emphasise the need to determine the possible biases introduced by storage and transport methods in all microbiome studies. Although immediate freezing of faecal samples at −80 °C preserves the microbiome composition when compared to the ‘gold standard’ of immediate DNA extraction, a recent study showed that while more variation was seen in samples stored at room temperature and in cool boxes, no significant differences were found between groups stored at room temperature, in cool boxes, in a −80 °C freezer for most bacterial populations28. As shown by many microbiome studies, the variation between participant samples is often greater than that related to differences in storage.

While the main aim of this study was not to define a core microbiome in healthy TB racehorses, we have described the baseline microbiome composition in a moderate number of horses of a similar age kept under similar conditions and who were free of overt metabolic or intestinal disease and injury at the start of the clinical trial. The baseline microbiome of the trial horses more closely resembled that found in horses with colitis [1] than to what has been reported for healthy/forage fed/pasture grazed horses (Appendix A) [1,3,8]. This suggests/indicates that actively racing horses may have a faecal microbiome which is not in balance and may eventually lead to adverse health effects should the imbalance become more prominent. 

Due to the large between horse variability for relative abundances of the bacteria, it was only possible to show an overall effect of treatment median relative bacterial abundance of the phylum Bacteroidetes in this trial. However, this trial did show that the percent changes from baseline at 12 weeks were significantly different from zero for the treated horses. This highlights the reason the study was planned a priori to carry out a blinded assessment of change for each horse from its own baseline as a proof of concept in testing a supplement intended to enhance the intestinal microbiome. As only 71 to 84% of reads were classified to genus level, and even fewer were classified to species level, we did not take our analysis any deeper. Future longitudinal studies with a 12 to 36 month follow up and a more in-depth analysis of the microbiome in conjunction with the metabolome are required to identify important microbiome changes associated long-term supplementation.

## 5. Conclusions

The current study found that the prebiotic supplement resulted in faecal microbiome changes in the supplemented group to more closely resemble that seen in forage fed/pasture grazed horses rather than those with colitis. Microbiomes can be highly variable for numerous reasons and this study provides useful information for further investigations on long-term effects on the microbiome and on health and racing-related outcomes.

## Figures and Tables

**Figure 1 vetsci-10-00027-f001:**
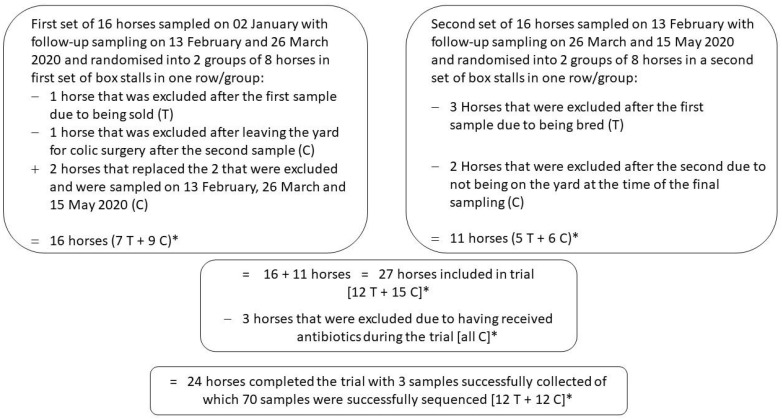
Flow chart of horses enrolled in the clinical trial. * Treatment group assignment was not revealed until after the outcome was assessed.

**Figure 2 vetsci-10-00027-f002:**
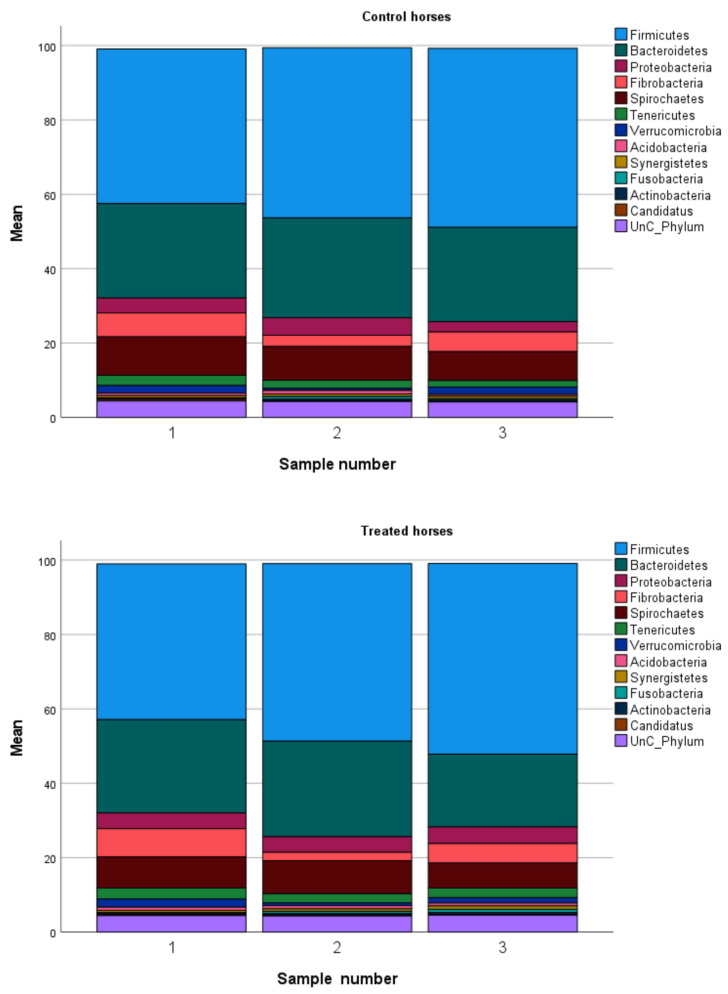
Microbiome composition showing mean relative abundances of 12 bacterial phyla and an unclassified category for 10 control horses and 12 treated horses given a prebiotic nutritional supplement at baseline and at 6- and 12-weeks.

**Table 1 vetsci-10-00027-t001:** Results of a pilot study of 4 horses supplemented for 2 months.

Bacteria	Baseline Samples					After 2 Months of Supplementation					% Change in Relative Abundance				
	Mean	Median	SD	Min	Max	Mean	Median	SD	Min	Max	Mean	Median	SD	Min	Max
Firmicutes:Bactoidetes Ratio	1.16	1.16	0.49	0.67	1.63	3.14	2.00	2.80	1.26	7.29	1.98	0.84	2.46	0.58	5.66
Phylum:															
*Firmicutes*	32.94	31.95	4.59	28.73	39.12	46.84	47.58	10.38	33.45	58.75	13.90	12.87	8.95	4.72	25.16
*Bacteroidetes*	31.69	31.73	10.22	20.59	42.69	20.66	23.97	8.65	8.06	26.65	−11.03	−12.08	5.11	−16.04	−3.91
*Spirochaetes*	7.19	7.39	4.75	1.63	12.35	7.77	8.17	2.27	4.78	9.97	−0.01	−2.30	5.74	−3.78	8.35
*Fibrobacteria*	2.4	1.52	2.53	0.44	6.11	0.10	0.10	0.04	0.04	0.14	−2.30	−1.40	2.56	−6.07	−0.33
*Proteobacteria*	9.18	10.19	3.17	4.64	11.69	9.17	9.04	3.36	5.62	12.99	0.58	1.75	4.02	−5.01	3.84
*Tenericutes*	1.34	1.40	0.5	0.76	1.81	0.68	0.71	0.09	0.54	0.74	−0.67	−0.77	0.49	−1.12	−0.02
*Verrucomicrobia*	3.77	2.37	3.47	1.42	8.93	6.76	7.04	2.28	4.06	8.88	2.99	3.86	4.83	−3.24	7.46
*Acidobacteria*	0.03	0.03	0.01	0.01	0.04	0.01	0.01	0.00	0.00	0.01	−0.02	−0.02	0.02	−0.04	0.00
*Synergistetes*	0.31	0.32	0.03	0.27	0.35	0.14	0.13	0.07	0.07	0.22	−0.18	−0.18	0.06	−0.24	−0.11
*Actinobacteria*	0.86	0.91	0.34	0.41	1.22	0.69	0.68	0.28	0.39	1.03	−0.17	−0.11	0.56	−0.83	0.38
*Fusobacteria*	0.004	0.002	1.07	5.64	8.24	0.01	0.01	0.01	0.004	0.02	0.01	0.01	0.01	−0.01	0.02
*Candidatus*	0.000	0.000	0.000	0.000	0.000	0.000	0.000	0.000	0.000	0.000					
Unclassified phylum	7.06	7.19	0.005	0.000	0.01	4.93	4.88	0.38	4.52	5.43					
Class:															
*Clostridia*	28.37	27.51	4.07	24.4	34.06	42.35	42.33	9.09	31.28	53.46	13.98	11.40	8.62	6.88	26.24
*Erysipelotrichia*	1.02	0.75	0.83	0.36	2.23	1.71	1.72	0.80	0.73	2.68	0.69	0.63	1.16	−0.48	1.99

**Table 2 vetsci-10-00027-t002:** Descriptive statistics showing the mean, median, standard deviation (SD), minimum (Min) and maximum (Max) values for the relative abundances of bacteria as a proportion of the total bacterial counts at baseline and 12 weeks with the percent change from baseline for 10 control horses and 10 treated horses. Significant differences between control and treated horses (within columns) are highlighted in bold. Significant differences for percent change from baseline to the 12-week sample being significantly different from zero are highlighted with symbols and in footnote.

10 Control Horses															
Bacteria	Baseline Samples					after 12 Weeks of Supplementation					% Change in Relative Abundance				
	Mean	Median	SD	Min	Max	Mean	Median	SD	Min	Max	Mean	Median	SD	Min	Max
SSD	2.82	2.82	0.12	2.67	3.04	2.90	2.97	0.23	2.41	3.14	3.24	4.14	11.19	−16.87	17.67
Number of Species	**697.20 ***	696.00	68.06	617.00	805.00	865.40	898.50	87.02	747.00	961.00	**24.53 ^##^°**	**23.34 ^##^°**	11.14	4.85	42.46
*Firmicutes:Bactoidetes* Ratio	1.80	1.73	0.80	0.79	3.38	2.20	1.79	1.40	0.81	5.99	33.28	14.95	63.56	−63.03	140.61
Phylum:															
*Firmicutes*	41.61	42.33	9.95	26.20	55.12	48.16	47.22	8.53	36.47	63.29	20.79	19.31	29.50	−28.76	66.89
*Bacteroidetes*	25.37	25.60	6.69	16.31	36.75	25.34	**24.53 ^#^**	8.29	10.57	44.92	4.30	−13.11	41.08	−35.19	92.71
*Spirochaetes*	10.33	9.29	5.26	4.81	24.10	7.77	6.18	5.16	1.88	17.91	−2.20	−50.02	89.64	−78.83	172.14
*Fibrobacteria*	6.45	3.74	6.79	1.42	23.90	5.29	4.09	4.47	1.25	16.33	68.21	−27.37	188.67	−86.57	466.89
*Proteobacteria*	4.02	3.75	0.84	3.06	5.70	2.81	2.26	1.82	1.43	7.46	−24.57	−50.82	56.05	−74.91	112.54
*Tenericutes*	2.76	2.99	1.26	0.46	4.70	1.76	1.77	0.73	0.39	2.85	−1.08	−35.05	112.17	−89.46	304.35
*Verrucomicrobia*	2.06	0.99	2.15	0.23	7.24	1.97	0.93	2.65	0.36	9.12	17.67	−5.72	84.79	−72.65	164.35
*Acidobacteria*	0.66	0.68	0.13	0.49	0.89	0.48	0.33	0.37	0.18	1.31	−25.05	−48.89	56.28	−74.33	74.67
*Synergistetes*	0.61	0.45	0.45	0.00	1.41	0.67	0.54	0.36	0.25	1.37	20.75	10.47	68.04	−72.48	147.31
*Fusobacteria*	0.07	0.04	0.05	0.03	0.16	0.41	0.13	0.85	0.01	2.80	844.31	103.13	2095.05	−95.97	6729.27
*Actinobacteria*	**0.28 ****	0.25	0.08	0.21	0.49	0.35	0.33	0.12	0.22	0.60	36.93	27.46	63.90	−53.47	147.92
*Candidatus*	0.45	0.38	0.31	0.09	1.23	0.13	0.10	0.08	0.04	0.31	−57.18 °	−77.51 °	39.39	−93.06	35.11
Unclassified_phylum	4.43	4.42	0.95	3.12	6.46	4.14	3.59	1.55	2.65	7.44	−0.78	−20.08	54.00	−44.91	138.54
Class:															
*Clostridia*	29.50	31.70	6.29	20.54	37.72	37.42	36.19	10.76	22.96	59.19	31.98 °	39.52 °	40.98	−36.39	82.06
*Erysipelotrichia*	6.73	6.70	3.79	1.59	14.34	4.80	4.46	2.92	1.50	11.34	13.81	−20.65	105.33	−87.71	196.91
**10 Treated Horses**															
SSD	2.87	2.91	0.16	2.60	3.10	2.97	2.97	0.09	2.83	3.15	3.57	3.83	5.07	−4.17	9.61
Number of Species	**801.40 ***	791.50	89.48	687	934	843.10	817.50	168.99	590	1098	**5.88 ^##^**	**−0.52 ^##^**	22.36	−22.47	44.10
*Firmicutes:Bactoidetes* Ratio	1.77	1.85	0.63	0.77	3.11	3.14	2.68	1.54	1.37	6.02	84.74 °°	70.79 °°	87.19	−4.23	295.36
Phylum:															
*Firmicutes*	42.36	44.66	6.83	29.49	50.70	51.16	50.39	6.86	41.37	64.94	22.85 °°	20.19 °°	20.19	−2.06	63.58
*Bacteroidetes*	25.75	23.89	6.47	16.29	40.41	19.05	**19.31 ^#^**	6.53	8.88	30.20	−24.57 °°	−24.93 °°	22.43	−70.31	3.21
*Spirochaetes*	8.54	8.37	2.43	5.45	13.42	6.98	7.47	4.00	1.48	13.64	−17.44	−31.14	49.81	−76.91	58.24
*Fibrobacteria*	6.63	5.62	4.80	1.32	14.11	5.82	2.56	7.44	0.98	23.72	54.00	−52.46	209.45	−85.33	531.36
*Proteobacteria*	3.87	3.12	1.92	2.01	7.51	4.42	3.32	2.78	1.62	9.75	25.05	3.55	72.36	−54.14	152.59
*Tenericutes*	3.02	3.21	1.16	1.60	5.30	2.47	1.67	2.75	0.12	9.48	−23.68	−39.86	75.14	−94.28	171.63
*Verrucomicrobia*	2.19	1.92	1.46	0.64	4.76	1.65	1.33	1.30	0.26	3.77	20.25	−21.23	110.29	−94.45	180.62
*Acidobacteria*	0.74	0.57	0.53	0.36	2.17	0.75	0.57	0.48	0.24	1.63	31.38	−10.42	102.95	−72.26	251.29
*Synergistetes*	0.61	0.63	0.18	0.29	0.93	0.96	0.87	0.37	0.58	1.68	72.03	12.57 *	113.41	−18.05	323.79
*Fusobacteria*	0.07	0.07	0.04	0.02	0.17	0.53	0.14	0.71	0.01	1.82	1252.22	130.01	2124.69	−92.35	5725.00
*Actinobacteria*	**0.38 ****	0.37	0.11	0.21	0.56	0.51	0.42	0.29	0.30	1.25	36.53	8.29 °°	57.12	−24.69	150.00
*Candidatus*	0.38	0.33	0.16	0.19	0.66	0.17	0.13	0.21	0.00	0.73	−29.11	−55.17	113.81	−99.26	284.21
UnC_Phylum	4.47	4.62	0.81	3.18	5.58	4.61	4.48	1.13	3.17	6.94	6.77	4.36	32.51	−40.50	54.65
Class:															
*Clostridia*	30.89	32.95	8.08	20.26	45.60	42.02	41.69	8.63	29.88	59.98	40.24 °°	35.61 °°	25.86	−8.18	78.54
*Erysipelotrichia*	5.96	4.93	2.95	2.29	13.03	3.40	3.80	1.69	0.37	6.01	−37.28 °°	−37.99 °°	33.61	−94.12	28.78

* *p* = 0.009 for mean and *p* = 0.018 for median number of species in control horses compared to treated horses. ** *p* = 0.04 for mean and *p* = 0.018 for median abundance of *Actinobacteria* in control horses compared to treated horses. ^#^ *p* = 0.08 for mean and *p* = 0.04 for median abundance of *Bacteroidetes* in control horses compared to treated horses. ^##^ *p* = 0.03 for mean and *p* = 0.02 for median percent change in the number of species in control horses compared to treated horses. ° *p*-values for percent change from baseline to the 12-week sample being significantly different from zero for the control horses: *p* < 0.001 for mean and *p* = 0.005 for median percent increase in species, *p* = 0.001 for mean and *p* = 0.009 for median percent decrease in *Candidatu*s, *p* = 0.036 for mean and *p* = 0.047 for median percent increase in *Clostridia.* °° *p*-values for percent change from baseline to the 12-week sample being significantly different from zero for the treated horses: *p* = 0.013 for mean and median percent increase in *Firmicutes: Bacteroidetes* ratio, species, *p* = 0.006 for mean and *p* = 0.007 for median percent increase in *Firmicutes*, *p* = 0.007 for mean and *p* = 0.013 for median percent decrease in *Bacteroidetes*, *p* = 0.028 for median percent increase in *Actinobacteria*, *p* < 0.001 for mean and *p* = 0.007 for median percent increase in *Clostridia* and *p* = 0.007 for mean and *p* = 0.013 for median percent decrease in *Erysipelotrichia*.

**Table 3 vetsci-10-00027-t003:** Crosstabulation of two definitions of outcome for changes in the faecal microbiome from baseline after 12 weeks of supplementation with criteria for change and measures of effect. Significantly more treated horses showed changes compared to control horses.

**Group**	**1st Definition of Change ***		**Total**
	**Yes**	**No**	
W = Treatment	8	2	10
X = Control	2	8	10
Total	10	10	20
**Group**	**2nd Definition of Change ****		**Total**
	**Yes**	**No**	
W = Treatment	8	2	10
X = Control	0	10	10
Total	10	10	20

* Criteria for 5 changes in 1^st^ definition: Phylum *Firmicutes* (F) ≥ 9% increase. Class *Clostridia* (Cl) ≥ 24% increase. Phylum *Bacteroidetes* (B) ≥ 5% decrease. F:B ratio ≥16% increase. Phylum *Actinobacteria* ≥ 2% increase. Measures of effect for 1^st^ definition: Control Event Rate (CER) = 0.2 = 20%. Experimental Event Rate (EER) = 0.8 = 80%. Relative risk (RR) = 4.0 (95% CI: 1.1–14.4, P=0.012).** Additional criterion for 2^nd^ definition: Class Erysipelotrichia of ≥8% decrease. Measures of effect for 2^nd^ definition: Fisher’s exact *p* = 0.0004. Control Event Rate (CER) = 0. Experimental Event Rate (EER) = 0.8. Relative risk (RR) = 17.0 (95% CI: 1.1–259.9, P=0.0004).

## Data Availability

The data that support the findings of this study are available from the corresponding author upon reasonable request.

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
