# Peer review of "Results of a Clinical Trial Showing Changes to the Faecal Microbiome in Racing Thoroughbreds after Feeding a Nutritional Supplement"

_vetsci, 2022, doi:10.3390/vetsci10010027_

Round 1
Reviewer 1 Report
Microbiome changes after a nutritional supplement
Line 5 Authors- there is an extra Vicki J after Adams
Simple Abstract: please add: length of time on supplement,
Scientific abstract: type of supplement more specific then just prebiotic (what type of plants? Gum acacia? Ratio of Fructo-oligosaccharides vs galacto-oligosaccharides etc, where manufactured? QC assurances? Other fillers/additives) maybe not all can be in abstract but please include in paper
67 h in front of forage to be removed please
101 Did you have Hay and grain analysis to look at differences between yards? Did the hay change in the middle of the study? Were all horses on the same type of hay/grain?
126 did you control for the time of year? As second set were into spring?
169- did the PI know that the horses were assigned in a block fashion?
210 what tests were used to assess normality?
211 why were t tests chosen when you have 3 time points and thus a repeated measures design?
Why was the 6 week time excluded?
218 why did you do both parametric and non-parametric tests on the same data?
229- how did you determine these % were clinically significant? Refs?
237 what areas had N<5 that you chose this over a Chi Square?
Where are any of your results from the 6 week time point? Why did you include this in your paper if you are only analyzing 12 weeks?
253- how long were the horses on this diet before the baseline sample? 6 weeks or? This is important so we know the change isn’t due to a diet change from a prev farm. Please include in methods how long horses had been on the diet before baseline sampling.
269 remove extra period
273 for the castrated colts and actually for any of the horses, how was it ensured that they ate the supplement? Did any leave any food or not eat the supplement? Was the amount of food left uneaten weighed?
274-277 this sentence about the food types is confusing, it looks like only one type is outside of the () please clarify type 1 is X and type 2 is Y. what is RED mill nutrition care package?
277 were hay and haylage fed in similar amouts to each horse or were some on hay and others on haylage? Were those analyzed? Were the same batches of hay and haylage fed throughout the trial and were they the same from first first group to the second? If not, was this controlled for?
281 were these horses in full training even though they didn’t race?
288 was the gelded horses not given any anti-inflammatories???? Did he go off feed at all?
The table after line 318 is cut off on the right
The horses that did race, how often did they race (separated by group)
In discussion, need more references to many equine papers where probiotics didn’t help and some analysis of why prebiotics made a change, need more here
Id remove reference to 6 weeks exams if they are never discussed. I am interested to hear what the analysis showed however as 6 weeks is a common endpoint for many studies, so if not significant then it can help shape future protocols.
Reviewer 2 Report
Dear authors,
Line 23-24: This provides useful information for further investigations on long-term effects on the microbiome and on health and racing-related outcomes.
It is not clear "racing related outcomes" of?
Lines 123-124: The horses in the treatment group received one heaped scoop of the pelleted nutritional supplement (~10 grams) once daily for 12 weeks.
Is this correct? 12 grams of supplement only? for a 550kg horse?
-Material and methods are extensive in my opinion. By shortening them and making them more factual, the reading of the paper will be easier. Additionally the data on the horses (age, sex, breed, etc) should be stated here and not after in the results/discussion.
-Table A10 is incomplete, it cannot be seen fully
Lines 478-479 "The pilot study baseline samples were collected in the summer (June 2018) and the microbiome showed greater similarity to that of the horses with colitis in the study by Costa and others than the trial horses did at baseline [1]." This is not clear. Why compare with colitis on normal horses? are the control horses with disbiosis of the intestinal microflora?
Lines 529-530: The comment also follow the previous questioning, further, the definition of the micriobiome of a horse with colitis is not given.
The horses of this study did not have colitis, or the prevalence of colitis within the group was not studied or analyze, was it compared with sick horses?
General comment:
Overall the article is interesting and a prove on how doable is to improve the microflora in horses that live on stable/stressful conditions. The scientific study is well developed and represented in the manuscript.
